# Updates in Glioblastoma Immunotherapy: An Overview of the Current Clinical and Translational Scenario

**DOI:** 10.3390/biomedicines11061520

**Published:** 2023-05-24

**Authors:** Andrea Bianconi, Giuseppe Palmieri, Gelsomina Aruta, Matteo Monticelli, Pietro Zeppa, Fulvio Tartara, Antonio Melcarne, Diego Garbossa, Fabio Cofano

**Affiliations:** 1Neurosurgery, Department of Neurosciences, University of Turin, 10126 Turin, Italy; gel.aruta@gmail.com (G.A.); pietro_zeppa@yahoo.it (P.Z.); amelcarne@cittadellasalute.to.it (A.M.); diego.garbossa@unito.it (D.G.); fabio.cofano@gmail.com (F.C.); 2Neurosurgery, Ospedale SS. Annunziata, 74121 Taranto, Italy; giuseppe.palmieri@edu.unito.it; 3UOC Neurochirurgia, Dipartimento di Medicina Traslazionale e per la Romagna, Università degli Studi di Ferrara, 44121 Ferrara, Italy; mmonticelli89@gmail.com; 4Headache Science and Neurorehabilitation Center, IRCCS Mondino Foundation, Department of Brain and Behavioral Sciences, University of Pavia, 27100 Pavia, Italy; tartarafulvio@gmail.com; 5Humanitas Gradenigo, 10100 Turin, Italy

**Keywords:** glioma, glioblastoma, immunotherapy, peptide vaccines, CAR-T therapy, oncolytic virus, oncolytic vaccines, immune checkpoints

## Abstract

Glioblastoma (GBM) is the most common and aggressive central nervous system tumor, requiring multimodal management. Due to its malignant behavior and infiltrative growth pattern, GBM is one of the most difficult tumors to treat and gross total resection is still considered to be the first crucial step. The deep understanding of GBM microenvironment and the possibility of manipulating the patient’s innate and adaptive immune system to fight the neoplasm represent the base of immunotherapeutic strategies that currently express the future for the fight against GBM. Despite the immunotherapeutic approach having been successfully adopted in several solid and haematologic neoplasms, immune resistance and the immunosuppressive environment make the use of these strategies challenging in GBM treatment. We describe the most recent updates regarding new therapeutic strategies that target the immune system, immune checkpoint inhibitors, chimeric antigen receptor T cell therapy, peptide and oncolytic vaccines, and the relevant mechanism of immune resistance. However, no significant results have yet been obtained in studies targeting single molecules/pathways. The future direction of GBM therapy will include a combined approach that, in contrast to the inescapable current treatment modality of maximal resection followed by chemo- and radiotherapy, may combine a multifaceted immunotherapy treatment with the dual goals of directly killing tumor cells and activating the innate and adaptive immune response.

## 1. Introduction

Glioblastoma multiforme (GBM) is the most common and aggressive type of primary intrinsic glial brain tumor in the adult population [1]. The current standard of care for patients with GBM involves surgical intervention, aimed at achieving gross total resection (GTR), followed by chemo–radiotherapy treatment [2,3]. Surgery remains the initial step in GBM management because GTR impacts the overall survival (OS); thus, the resulting performance status greatly influences subsequent treatment [4]. Despite significant improvements in both surgical techniques and intraoperative technology, with the introduction of Fluorescence Guided Surgery (FGS), intraoperative Magnetic Resonance Imaging (iMRI), and intraoperative Ultrasounds (iUS), prognosis for GBM remains bleak [5,6,7]. 

Although GBM rarely metastasizes, its malignant behavior, infiltrative growth pattern, and intrinsic and extrinsic heterogeneity make it one of the most challenging tumors to treat. After medical treatment, GBM has a median OS of 15 months, with only 5% of patients surviving beyond 5 years [2]. For patients with GBM relapses and disease progression, available treatments such as surgery, radiotherapy, and drug therapy with alkylating agents or bevacizumab guarantee an average life expectancy of about 6–9 months [8,9,10]. 

Cancer immunotherapy represents a novel treatment approach that manipulates the patient’s adaptative and innate immune response to fight off the neoplasm. Various immunotherapeutic approaches have shown promising results in improving prognosis and life expectancy for cancers outside of the brain, such as melanoma and leukemia [11,12]. Given these impressive results, immunotherapy was also studied in brain tumors [13,14,15]. It is worth noting that brain was once considered an immunologically distinct and privileged site protected by the blood–brain barrier (BBB), lacking specific antigen presenting cells (APCs), and closed off to circulating lymphocytes. However, this concept has been refuted over the last two decades due to the discovery of a lymphatic system in the brain that runs parallel to venous sinuses, as well as the ability of microglia/macrophage cells (M/M) to present antigens and activate lymphocytes following BBB breakdown [16,17]. 

Non-neoplastic cells are estimated to account for up to 30% of the GBM volume and are responsible for the development of the so-called tumor microenvironment, comprising glioma stem cells, stromal cells including resident microglial cells, and immune cells such as monocytes, tumor-associated macrophages, and lymphocytes [18,19]. The activation of microglial cells and their subsequent recruitment of circulating monocytes in the brain are primarily driven by chemokines, neurotransmitters, complement receptor ligands, and extracellular vesicles [20].

Along with the rapid development of the -omic era, research into immunotherapy has led to a deeper understanding of GBM genesis and the underlying mechanisms of the immunosuppressive environment that make GBM a “cold tumor” [21,22]. This concept is in contrast with the so-called “hot tumors”, the ones showing signs of local inflammation and that are already infiltrated by T lymphocytes. They are therefore ideal candidates for immune checkpoint inhibitors. In contrast, cold tumors need to be made “hotter”, promoting T cell infiltration and microinflammation, in order for these therapies to play a significant role. To date, immunotherapeutic approaches represent the most concrete and promising field of research in the fight against GBM. The aim of this review is two-fold: 1) to highlight the most frequently impaired molecular patterns in GBM genesis and the innate and acquired mechanisms of immune resistance, and 2) to explore the development of targeted immunotherapeutic strategies that may be effective against these impairments. 

## 2. Immunotherapy Hints in Glioblastoma Treatment

As an immunosuppression-associated tumor, GBM negatively affects the local and systemic immune system’s response to both tumor antigens and intrinsic factors [23,24]. While the exact mechanism of these effects remains unknown, numerous theories have been proposed over time. Some of them suggest that various signaling pathways induced by GBM lead to immunosuppression [25,26]. In order to understand GBM immunosuppressive pathways, we must first say that the tumor microenvironment is made up of a mixture of two different types of cells: microglia (CNS-resident macrophages) and monocytes, which originate from hematopoietic stem cells and travel through the blood to reach the brain when the blood–brain barrier is damaged. 

In particular, the overexpression of STAT3 signaling in human GBM is responsible for an IL-10-induced immunosuppression. Glioma-associated macrophages are known as the main source of IL-10 in human GBM, but tumor cells are also responsible for the production and secretion of this interleukin. In fact, IL-10 was one of the first immunosuppressive soluble factors whose level was found to be increased in GBM patients [26]. Additionally, transforming growth factor beta (TGFβ), a peptide involved in T cell activity inhibition, is also believed to play a crucial role in immunosuppression-associated tumors, as it was found to be secreted by microglia cells in such tumors [14]. 

Two additional inhibitory immune pathways, programmed cell death protein 1 and its ligand (PD-1 and PD-L1) and cytotoxic T lymphocyte associated antigen 4 (CTLA-4), have been identified as significant barriers to GBM patients’ immune response [27,28]. Increased levels of PD-L1 have been found in macrophages isolated from GBM patient blood [29], as well as CTLA-4, a classic regulatory feedback inhibition model that downregulates the amplification of T cell responses [30,31].

GBM also evades the immune system through the direct interaction between GBM cells and immune system cells. GBM suppresses natural killer (NK) cell activity by inducing apoptosis via tumor necrosis factor receptor superfamily member 6 (TNFRSF6) or, again, PD-L1 [32,33,34]. 

Interestingly, patients with GBM exhibit reduced T cell expression compared to other types of tumors, such as melanoma and breast cancer, where a robust T cell response is often observed [12]. This phenomenon of T cell exhaustion renders GBM patients less responsive to immune checkpoint blockade (ICB). The exact cause of this T cell depletion in GBM is not fully understood, although some studies suggested a possible association with the immunosuppressive M2-like phenotype of GBM myeloid cells [35,36,37]. Other animal studies suggest that the ingression of myeloid cells, such as dendritic cells (DCs), from the periphery is necessary to elicit an immune response, which is absent in GBM patients [38]. 

Immune checkpoint (IC) inhibition, specifically targeting the PD-1/PD-L1 and CTLA-4 inhibitory pathways, has become a primary focus of GBM-induced immunosuppression research [27,33,34]. Monoclonal antibodies against PD-1 (e.g., nivolumab and pembrolizumab), PD-L1 (e.g., atezolizumab and durvalumab), and CTLA-4 (e.g., ipilimumab) have been extensively tested in clinical trials, either alone or in combination with other drugs [39,40,41]. However, evidence for their benefits in terms of OS is not yet clear [42,43]. In addition to IC inhibition, research in antitumor vaccination, including peptide and DC vaccines, chimeric antigen receptor (CAR) T cell therapy, and oncolytic viral therapy is also gaining momentum in the field of GBM immunotherapy (Figure 1) [14,22]. 

## 3. Immune Checkpoint Inhibitors

In a normal immune system, molecules involved in the so-called immune checkpoint (IC) are of capital importance in reducing abnormal T cell activity and therefore in avoiding autoimmune disorders [44,45]. Some of these molecules have a “positive” activating effect on the immune system, such as CD28, a type of co-stimulatory molecule which is expressed in about 90% of CD4+ T cell and 50% CD8+ T cell, that upregulates the effector of T cell activator [46]. In reverse, some others can present a “negative” effect on the immune system. CTLA-4, also known as CD152, for instance, blocks co-stimulatory signals binding specific factors [28]. In particular, CTLA-4 is not constitutively expressed on the T cell surface, like other ligands, and is only found in the activated conventional T cells and CD4+Foxp3+ regulatory T cells. T cell activation could be suppressed by CTLA-4 in an antigen-specific way, by interrupting co-stimulatory signaling and functioning as an inhibitor of naïve and memory T cells [47]. In addition, immune reactivity itself could be downregulated by CTLA-4 with the reduction of helper T cells (Th) (Figure 2).

Some neoplastic cells, APCs, B cells, as well as parenchymal cells, express on their surface programmed death ligand 1 (PD-L1). This peculiar ligand induces T cell apoptosis, binding to its receptor programmed death 1 (PD-1) that is present on activated T cells mainly in peripheral organs. PD-L1 expression can be augmented by inflammatory cytokines, particularly interferons, and at the same time PD-L1 promotes CD8 + T cell production of tumor-specific interferon-γ [48]. PD- L1 is expressed in multiple tumors, including glioblastoma [34,49]. In GBM, expression of PD-L1 on the surface of tumor cells has been linked to the phosphatase and tensin homolog (PTEN) loss and PI3K-PTEN-AKT-mTOR signaling pathway overactivation [50]. Thus, multiple aspects of immune reactivity can be enhanced by the therapeutic targeting of PD-1 associated with Tregs, cytotoxic T cells, B cells, and NK cells [14,22]. 

Furthermore, there is interest in checkpoints expressed in other immune cell populations, such as NK cells. Delconte et al. found that the suppressor of cytokine signaling (SOCS) family member, cytokine-inducible SH2-containing protein (CIS), functions as a crucial intracellular negative regulator of activated NK cells [51]. More importantly, they showed that CIS blockage increases antitumor activity of NK cells. The authors also found that the combination of CIS inhibition with CTLA4 and PD1 blockade had a greater effect in reducing melanoma metastasis than either of these treatments alone. CIS inhibition may offer an alternative therapeutic option for patients who failed with other immune checkpoint inhibitors [51]. The potential for NK-targeted agents to augment the antitumor effects of a T cell checkpoint blockade is actively under consideration. A number of promising NK-targeting therapeutics are in early phase trials (Table 1) [52]. 

As aforementioned, cancer cells can exploit immune checkpoints to evade immune attack and suppress immune destruction. Preclinical trials, as well as various stages of clinical trials, have proved the efficacy and safety of several types of immune checkpoint inhibitors [41,53,54,55]. In particular, CTLA-4 and PD-1 inhibitors have been studied in depth, with the demonstration of exciting results in clinical cancer therapy. Ipilimumab, known as a fully humanized IgG1 subclass monoclonal antibody (mAb) against CTLA-4, demonstrated significant antitumor power while other conventional therapies for metastatic melanoma remained dismal. It was approved for melanoma therapy by the FDA in 2011 and became part of routine melanoma treatments [56,57,58,59]. Another humanized anti-CTLA-4 antibody, tremelimumab, obtained durable responses in phase I/II clinical studies in melanoma, but failed in a phase III randomized clinical trial [60,61,62,63]. 

The PD-1/PD-L1 axis has also been shown to be a potential target in tumor tissues [43,64]. PD-1 inhibitor nivolumab has been proven to extend OS and PFS in melanoma patients [65,66]. Significant efficacy by the anti-PD-1 antibody was observed in ∼20–25% patients with both melanoma and lung and renal cancer [67,68,69]. In addition, the association between response to anticancer treatment and tumor PD-L1 expression before treatment has been observed in early clinical trials but may initially be achieved only in combination with certain vaccines [27]. In September 2016, the United States approved an anti-PD-1 drug, pembrolizumab, as treatment in metastatic melanoma after standard treatment [70,71,72]. Additionally, in a 2015 study nivolumab achieved a significant objective response rate (87%) in relapsed or refractory Hodgkin’s lymphoma [67]. 

As for the role in glioma treatment of immune checkpoint PD-1 and CTLA-4 targeting drugs, for a long time we believed that such drugs could not be used in brain cancer because of the impossibility of reaching glioma cells and microenvironment [73]. In fact, monoclonal antibodies are not believed to penetrate an intact BBB due to their large molecular size (150 kDa) [42]. However, several phase II and III trials demonstrated a certain efficacy of ipilimumab and nivolumab in the treatment of melanoma brain metastases [59,66,74]. 

It is believed that the pharmacodynamics effect of immune checkpoint inhibitors on brain tumors is caused in part by their activity on peripheral T cells, which then cross the BBB and act against cancer cells [42], and partly because both nivolumab and ipilimumab can indirectly enter the brain. They are, indeed, IgG monoclonal antibodies with FcRn binding, which can enter cells like macrophages in the choroid plexus and so reach the CSF (cerebrospinal fluid) via endocytosis via FcRn-mediated transcytosis [32,75]. 

Several studies demonstrate that immune checkpoint inhibitors could bring promising benefits for patients with GBM. In 2014, the first large phase III trial of nivolumab plus ipilimumab vs. bevacizumab in recurrent glioblastoma (NCT02017717) was initiated [76]. In the last update, the final endpoint was not reached, with a median overall survival of 9.8 months with nivolumab versus 10.0 months with bevacizumab [77,78]. In a recently concluded study, CheckMate 498 (NCT02617589), investigators explored nivolumab as an alternative to temozolomide TMZ (both in combination with radiotherapy) in patients with MGMT-promoter-unmethylated tumors [79]. The median OS in the nivolumab patients was 13.40 months, while in TMZ was 14.88. The ongoing study CheckMate 548 (NCT02667587) is evaluating nivolumab as an addition to the standard TMZ/RT→TMZ regimen in patients with MGMT-promoter-methylated tumors. In the last update given, median PFS was 10.64 months with radiotherapy, temozolomide, plus nivolumab versus 10.32 months in radiotherapy, temozolomide, plus placebo [80]. In a 2020 study, a significant improvement in survival was noted in both the wild type and the CD73−/− GBM-bearing mice that were treated with a combination of anti-PD-1, compared to controls [81]. A combination of nanotechnology and immunotherapy led Galstyan et al. to deliver targeted nanoscale immunoconjugates (NICs) on a natural biopolymer scaffold, as well as on poly (β-L-malic acid), with covalently attached a-CTLA-4 or a-PD-1 for systemic delivery across the BBB, with a local immune system activation and a prolonged survival in GBM-bearing mice [82]. Several ongoing clinical studies are testing the safety and toxicity of ipilimumab and nivolumab in glioblastoma patients, further exploring their efficacy in glioma treatment [39,43], but given the not-so-brilliant results obtained so far, their routine application in clinical practice seems still not indicated. 

## 4. Peptide Vaccines

Synthetic peptide vaccines are subunit vaccines, usually composed of about 20–30 amino acids. They contain a specific epitope of a tumor-associated antigen that triggers direct or potential immune responses, inducing an effective antitumor T cell reaction [83]. While many tumor antigens have been discovered to be presented in GBM, only a restricted number of them have been evaluated as potential targets for peptide vaccines. Proteins frequently mutated or atypically expressed in GBM include EGFR, NF1, PDGFRA, PTEN, TERT, RB1, TP53, IDH1, PIK3CA, and PIK3R1 [83]. Because of the large heterogeneity of tumor antigens, immunoresistance to a vaccine for a single antigen is common, so currently the trend is to explore vaccines that target multiple antigens; many running clinical trials are testing multiple mixed protein complexes against several GBM subtypes. (Table 1) [14]. 

### 4.1. EGFR vIII Vaccine

EGFRvIII is a ligand-independent, constitutively active splice variant of EGFR that has been proved to promote tumor growth and resistance to ajuvant TMZ treatment [84,85]. It is expressed in about 30% of GBM [86,87]. The sequence that codes for EGFRvIII lies primarily on episomal bodies [88,89]. 

Rindopepimut (CDX-110) is an injectable peptide vaccine, constituted by a 14-mer peptide that specifically spans the length of EGFRvIII, conjugated to the non-specific carrier protein Keyhole Limpet Hemocyanin. At the beginning, rindopepimut was used to elicit the formation of EGFRvIII-specific monoclonal antibodies, which were shown to mediate effective antitumor responses against EGFRvIII-positive cells [90]. Pre-clinical studies on rindopepimut-vaccinated mice showed an increase in EGFRvIII-specific humoral response, which can suppress tumor growth and enhance survival [91,92]. In the phase I trial VICTORI, patients with first-diagnosis GBM underwent vaccination with rindopepimut-pulsed, monocyte-derived dendritic cells and data suggested that most patients developed an EGFRvIII-mediated immune activation with a very low adverse effect incidence. Even if the medication was generally well tolerated, the trial did not show a statistically significant improvement in OS and PFS [93]. The safety and efficacy of rindopepimut in EGFR vIII-positive GBMs is now under examination in three phase II trials: ACTIVATE I, vaccination alone, and ACT II and ACT III vaccination in combination with adjuvant TMZ chemotherapy. In all of these trials, the vaccine has been administered with GM-CSF [85,90,94]. The results of these studies additionally validated the security of rindopepimut and proved a statistical improvement in median PFS and OS in vaccinated patients, compared with the cohort treated with standard therapy (PFS = 6.4 months, OS = 15.2 months) [85,90,94]. Other trials are now on the patients’ enrollment phase, and other studies are needed in order to improve our knowledge about this promising therapeutic option. 

### 4.2. IDH1 R132H Vaccine

Nowadays, it is well known that mutations in the isocitrate dehydrogenase 1 (IDH 1) enzyme are frequently present in gliomas; historically, the presence of an IDH gene mutation was the first to be associated with a better prognosis in glioblastoma, so much so that in the new WHO 2021 classification, its presence is sufficient for the diagnosis of low-grade glioma and excludes that of GBM. Specifically, the IDH 1 R132H mutation is present in among 6–10% of GBMs (according to WHO 2016 classification) and is typically associated with secondary GBMs that affect young adults [86,95]. IDH1 mutations are heterozygous and characteristically involve an amino acid substitution in the active site of the enzyme at codon 132. The mutation leads to the loss of normal enzyme function and abnormal production of 2-hydroxyglutarate (2-HG) [96]. 2-HG has been reported to suppress the enzymatic function of several alpha-ketoglutarate-dependent dioxygenases, among them histone and DNA demethylases, resulting in extensive modifications in histone and DNA methylation and thus possibly causing tumorigenesis [97,98]. As recently published by Schumacher et. al. in 2014, vaccination of MHC-humanized mice with a peptide vaccine presenting amino acids 123–142 (p123–142) of IDH1 R132H was able to suppress the growth of a sarcoma with IDH1 R132H-positivity. Along with the detection of MHC class II epitopes within p123–142, mice vaccinated with p123–142 possessed CD4 T cells reactive to IDH1 R132H and that yielded IFN-c. Additionally, CD4 depletion results in a reduction of tumor suppression due to vaccination [99]. Although the results from these papers are promising, additional studies are required to establish if a vaccine against IDH1 R132H is effective in the glioma setting, where typically MHC II is not expressed [100]. 

### 4.3. Cytomegalovirus Vaccine

Human cytomegalovirus (hCMV) is a common virus belonging to the herpesvirus family of which is extremely widespread globally. It is estimated that 40 to 80 percent of the population will encounter CMV infection, which usually evolves without symptoms and results in a latent infection [101]. Although those data are controversial for many reasons, it is a fact that various hCMV proteins were found in GBM samples and not in healthy tissues; these included IE1, US28, pp65, gB, HCMV IL-10, and pp28 [102]. Considering their exclusive presence in cancerous cells, these antigens have been indicated as immune-therapeutic targets. A clinical trial known as PERFORMANCE (NCT02864368) investigated the efficacy of a peptide vaccine, denominated PEP-CMV, containing both MHCI and II epitopes from CMV antigens. This trial was terminated in 2022 due to a lack of funding. However, preliminary results suggest that the vaccine may be capable of generating an immune response [103]. 

### 4.4. Dendritic Cell Vaccines

In Dendritic Cell (DC) vaccines, autologous DCs are activated ex vivo against specific glioblastoma-associated antigens, derived from tumour lysates, and subsequently reimplanted into the patient, in order to activate Cytotoxyc T- Lymphocytes (CTLs) through the major histocompatibility complex (MHC) class II-T cell receptor and CD80 or CD86–CD28 interactions. Activated CTLs have the capacity to recognize, and subsequently destroy, glioblastoma cells presenting specific antigens on the surface through the MHC class I proteins [104]. 

Different escape mechanisms have been found in GBM cells in destruction mediated by CTLs. One of these is represented by the suppression of lymphocyte activation through the up-regulation of immune checkpoint ligands, such as PD-L1; in fact, it can bind complementary receptors on the CTLs and reduce their action. Furthermore, the interactions between the CTLA-4 and CD80 and CD86 expressed on DCs surface prevent their binding with CD28, reducing the activations of CTLs by DCs. In the first case, an association of DC vaccines with monoclonal antibodies of immune checkpoint blockade can effectively prevent this interaction; similarly, an antibody-mediated blockade of CTLA-4 can be useful to prevent the second escape mechanism [94]. This reinforces the idea that a combination of different drugs with a synergic action could be the best therapeutic strategy in glioblastoma patients. 

One of the first attempts to evoke an immune response in GBM patients through DCs involved the vaccination of ten patients with GBM with autologous tumor lysate-loaded DCs after first-line therapy. An analysis of immune parameters measured before and after vaccination demonstrated that patients with an increase of at least one immune function parameter had improved survival. There were no serious adverse events related to DC vaccination [105]. Prins et al, in their prospective study, tried to find the best antigen combination for loading DCs in GBM patients. A total of twenty-eight patients were treated with autologous tumor lysate (ATL)-pulsed DC vaccination, while, because of HLA subtype restrictions on the associated antigen (GAA) peptide-pulsed DCs, only six patients were injected with GAA-DCs. In GAA-DC patients, compared with the ATL-DC patients, activated NK cells were found in higher frequencies. In addition, a significant correlation was observed between the decrease in the ratio of regulatory T lymphocytes and the overall survival of patients in either study [106]. 

The evidence that the recurrence and growth of some cancers is driven by cancer stem cells (CSC) leads to the idea of a DC vaccine targeting GBM CSCs [107]. Brain tumor biopsies were dissociated into single-cell suspensions, from which CSC-mRNA was amplified and transfected into monocyte-derived autologous DCs. These vaccines were injected intradermally only in patients in which corticosteroid therapy could safely be interrupted [108]. At the end, seven patients received the vaccine at specified intervals, after adjuvant radio- and chemotherapy. The immune response induced by vaccination has been proven in all patients, without any significant adverse effect. The comparison with controls showed that progression-free survival was 2.9 times longer in vaccine-treated patients [108]. 

ICT-107 is a multiple antigen DC-vaccine directed against six epitopes (AIM-2, MAGE1, TRP-2, gp100, HER2, and IL-13Ra2), and was tested in patients with a new diagnosis of GBM in a pilot study. It showed major activity against gp100 and HER2, with a prolonged mOS of 38 over a period of 4 months. The promising results led to a phase II trial that enrolled 124 patients with a new diagnosis of GBM, in which no improvement in OS was observed [109]. A phase III clinical trial with ICT-107 is scheduled to begin in December 2023 (NCT02546102). 

An autologous tumor-lysate vaccine called DCVaxL has been tested in combination with standard therapy in patients with a new diagnosis of glioblastoma, with the evidence of a prolonged mOS (23.1 months) compared to standard therapy alone (mOS 15–17 months); this vaccine has also been studied in combination with neo-antigen synthetic long-peptide vaccines, resulting in a mOS of 21 months [105,110]. A DC vaccine with fusions of DCs and glioma cells in combination with temozolomide (TMZ) chemotherapy was tested in a phase I/II trial in both patients with newly diagnosed GBM and those with recurrent GBM. The fusion cell immunotherapy proved that an antitumor response can be induced by acting against chemoresistance-associated peptides (WT-1, gp-100, and MAGE-A3) [111]. 

Evidence of human cytomegalovirus (CMV) in GBM patients’ brains has been found, leading to the creation of a vaccine in which DC cells were pulsed with cytomegalovirus phosphoprotein 65 (pp65) RNA. In this pilot study, 12 GBM patients were randomized to unilateral vaccine site pre-conditioning with unpulsed, autologous DCs or with a potent recall antigen such as tetanus/diphtheria (Td) toxoid, in order to increase lymph node homing and the efficacy of tumor-antigen-specific DCs [112]. Patients given Td had a higher accumulation of injected DC vaccines in their draining lymph nodes and a significantly improved survival. Years later, the same group demonstrated a favorable prognosis with an OS of 41.1 months in 11 newly diagnosed GBM patients injected with pp65-DCs following DI-TMZ [113]. 

Even if there is major evidence that the addition of DC vaccines to standard therapy is feasible and safe in glioblastoma patients and may extend overall survival, their preparation requires time and large costs. Moreover, though their biological activity has been largely demonstrated, clinical benefits are not always relevant enough. Certainly, more clinical trials are necessary in order to evaluate the potential survival benefit of DC vaccines, alone or in combinations with other immunotherapies [109,114]. 

## 5. CAR-T Therapy

Chimeric antigen receptor (CAR) T cell therapies are based on the infusion of engineered and manipulated T cells programmed to express chimeric antigen receptors (CAR) directed against specific, patient-tailored, tumor antigens [115]. Moreover, CAR T cells can recognize antigens that are not presented in the context of MHC-molecules and can be created already with an activated phenotype. All of these peculiar characteristics have made CAR T cells resistant to the immunosuppressive activity of the glioblastoma environment; therefore, they have aroused great interest within the scientific community [116,117]. CAR-engineered T cells have been recently used for CD19+ malignancies, such as acute lymphoblastic leukemia, chronic lymphocytic leukemia, and B cell lymphomas, providing durable and complete responses [11,118]. However, in many solid tumor studies, CAR T cell therapies induced an insufficient antitumor activity, maybe because targeting one antigen in a highly heterogeneous tumor might not be sufficient to eradicate all cancer cells [119,120]. In any case, all of these therapeutic strategies are drawn for patients with a limited tumor mass, mainly after surgical gross total resection, or as second-line treatment, in combination with other drugs [121]. 

EGFR variant III (vIII) is expressed in about 30% of newly diagnosed GBM, and represents, in patients surviving a year or longer, a negative prognostic indicator, regardless of other factors such as extent of resection and age [84,122]. EGFRvIII, targeted with a peptide vaccine strategy, has already been evaluated in phase II studies, while O’Rourke et al [123]. conducted a phase 1 study to evaluate the feasibility and safety of manufacturing and administering CART-EGFRvIII cells to patients with EGFRvIII-expressing, unmethylated MGMT (methylguanine-DNA-methyltransferase) promoter recurrent GBM. A total of 10 patients received a single dose of peripherally infused CART-EGFRvIII cells; each patient previously received second- and third-line treatments including surgery, bevacizumab, chemotherapy, or dendritic cell vaccine. The authors concluded as a primary end point that CART cell infusion was safe; in particular, no cytokine release syndromes were observed and only two patients required the administration of anti-IL6 therapy, although no clear correlation with a neurologic toxicity could be proved [123]. As a secondary end point, the authors observed a median OS of 251 days while PFS was not able to be determined because of confounding factors of neurosurgical intervention in most of the subjects. Only one patient remained alive without further therapy for more than 18 months after a single infusion of CART-EGFRvIII [123]. Interestingly, the study confirmed the CART-EGFRvIII cells’ engraftment in the peripheral blood and their trafficking in the brain with the reduction, in most of the subjects, of EGFRvIII expression, even though this could be the result of clone selection or antigen escape. Furthermore, the study of tumor microenvironment demonstrated in situ polyclonal T cell proliferation, possibly suggesting a secondary response by non-CAR expressing T cells. In fact, the phenotypic analysis demonstrated that many of these cells had an immunosuppressive function based on the expression of CD4, CD25, and Foxp3, suggesting the activation of an in situ compensatory multifactorial immunosuppressive response [123]. 

Brown et al. conducted a phase 1 study on CAR engineered T cells, targeting IL13alpha2 receptor (IL13Ralpha2) in three patients with recurrent GBM [124]. This pilot study was conducted on the basis of previous early phase clinical trials, and also considered that IL13Ralpha2 is overexpressed in more than 50% of GBM and not expressed at significant levels on normal brain tissue. Moreover, IL13Ralpha2 expression seems to be more closely associated with differentiated malignant cells and tumor infiltrating macrophages-derived suppressor cells, representing a prognostic indicator of poor patient survival [119,124]. The patients received multiple intracranial administrations of CART-IL13R-alpha2 following surgery via an implanted reservoir/catheter system. The authors were able to assess CART-IL13Ralpha2 safety and, at the same time, to observe encouraging evidence of transient anti-glioma activity with a mean survival of 11 months after relapse and best survival of 14 months. Interestingly, Brown et al. reported a single case of CART-IL13Ralpha2 intraventricular administration in a patient with diffuse GBM, with a complete response of intracranial and spinal lesions and a remarkable improvement of the quality of life for 7.5 months [124]. The authors speculate about the better efficacy of intraventricular administration in the case of leptomeningeal disease, also considering the significant increase in the CSF of interferon-gamma inducible chemokines with antitumoral potential after CART-IL13Ralpha2 infusion. Ahmed et al. conducted a phase 1 study to determine whether the systemic administration of HER2-specific CAR-modified virus-specific T cells (VSTs) was safe and whether these cells had anti-GBM activity on 17 patients with progressive HER2-positive GBM. As with the previous studies, the authors confirmed the safety and feasibility of this approach, of which could be associated with clinical benefit for patients with progressive GBM (OS: 11.1 months after T cell infusion) [125]. 

Despite these encouraging results, CAR-based strategies on a single molecular target are probably not sufficient to achieve a complete response in such a highly heterogeneous tumor, in which antigen escape and immunosuppressive pressure within the tumor still represent the main challenge to the fight against GBM [121]. The future direction of CAR-based strategies includes targeting multiple antigens, and in fact promising results in terms of mitigating antigen escape have been demonstrated with HER2 and IL13Rα2-directed tandem CAR-T cells, trivalent CAR-T cells targeting HER2, IL13Rα2, and EphA2 and CAR-T cells against EGFR and EGFRvIII in animal models [116,117,119]. 

## 6. Oncolytic Viruses

Anticancer therapies using the oncolytic viruses function are based on two distinct effects. In its first application, oncolytic viral therapy was used to exploit the virus’s ability to selectively infect and subsequently kill tumor cells. Later on, it was found that viruses can also activate the immune system through pathogen-associated molecular patterns, activate macrophages, and lure T cells into tumors, boosting the local immune response against tumor cells (Figure 3) [126]. Nowadays, replication-competent viruses, such as retroviruses, adenoviruses, herpes simplex viruses (HSVs), and polioviruses are used more and more, leaving behind initial concerns about the risk of encephalitis [127]. However, the future of oncolytic viral therapy seems to be tied to other immunotherapy strategies in combined approaches. 

The recombinant oncolytic poliovirus, PVSRIPO, is a genetically engineered form of the Sabin type I oral poliovirus, which exploits the natural neurotropism for the onco-fetal cell adhesion molecule (CD155), which is often upregulated in solid tumors, including GBM [128]. The implicated mechanism of action seems to be related to the GBM shifting from a cold to hot tumor, caused by the subversion of the innate antiviral interferon (IFN) response, resulting in viral cytotoxicity and antigen shedding. After a promising result in preliminary clinical data, PVSRIPO has more recently been evaluated on 61 patients with recurrent GBM, showing a mOS of 12.5 months with the 13% of the patients that remained alive at 3 years [104,128,129]. For these reasons, in May 2016 PSVRIPO received breakthrough therapy designation from the FDA. 

Adenoviruses represent a widely used target for anticancer therapies because they are a common respiratory virus and are easily manipulated in vitro for being engineered [130,131]. Two different kinds of engineered adenoviruses have been investigated: the first (DNX-2401) involves the manipulation of a viral capsid protein and has been evaluated in phase I trial in combination with Temozolomide, and after in phase II trials, in combination with anti PD-1 antibody (NCT02798406); results about the last one have not yet been disclosed to the public [132]. 

A similar phase I study, investigating DNX-2401 in combination with INF gamma (NCT02197169), was completed in 2018; based upon a preliminary intent-to-treat analysis, IFN did not appear to provide additional benefits to patients [133]. 

The second one (ONYX-015) involves the protein E1B, of which is implicated in the p53-induced cell apoptosis block and showed, in a phase I clinical trial, a mOS of 6.2 months. The low effectiveness of the study has been attributed to the high virus attenuation, necessary for keeping the safeness profile, of which might have negatively influenced the effectiveness of the viral action [134]. 

Adenoviruses have also been implicated in a different branch of immunotherapy, gene-mediated cytotoxic immunotherapy, in which the modified virus is used as a tumoricidal gene delivery vector. Adv-tk is a modified adenovirus transfected to express the Herpes Simplex Virus (HSV) thymidine kinase, which converts the ganciclovir in its active toxic nucleotide analogue form, of which kills replicating tumor cells. The safeness of this approach has already been evaluated in a phase I clinical trial, and two subsequent phase II trials have been conducted using intra-tumoral AdV-tk administration, valacyclovir or intraarterial AdV-tk administration and ganciclovir [135,136]. 

Cloughesy et al. investigated Toca 511 (vocimagene amiretrorepvec), a non-lytic retroviral replicating vector that delivers a yeast cytosine deaminase gene which converts the prodrug Toca FC into the antimetabolite 5-fluorouracil (5-FU) [137]. The selectivity for cancer cells is related to the high cell turnover that promotes viral spread and also to the lack of innate and adaptive immune responses that usually prevent genome integration [138,139]. In a phase I study, 45 subjects received, after tumor resection for recurrent or progressive HGG, intracavitary administration of Toca 511, followed by Toca FC intravenous administration. Toca FC represents an extended-release version of 5-fluorocytosine (5-FC) that easily crosses the blood brain barrier and is converted to 5-FU by the cytosine deaminase in the cancer cells infected by Toca 511 [16,140]. 5-FU can also act on non-infected cancer cells by direct diffusion through the cellular membrane [141]. Moreover, Toca 511 and Toca FC stimulate a local and systemic immune response that seems to contribute to the observed mOS of 13,6 months compared with historical controls [138,142]. For this promising result, a randomized phase II/III clinical trial for Toca 511 and Toca FC was scheduled in 2015, but unfortunately it was terminated by sponsor decision in 2020 without disclosure of the results [143]. 

HSVs have been extensively investigated in preclinical and phase I clinical trials with different genetic manipulations that include thymidine kinase deletion, γ134.5 dual knockout, viral ribonucleotide reductase disruption, and lacZ gene insertions into the viral ribonucleotide reductase gene promoter, concomitantly with processes that make the virus selective for tumor cells in the brain [126]. Different phase I studies and ongoing trials (NCT00028158, NCT00157703, NCT02457845, NCT02031965, NCT02062827) have shown antitumor effects and low neurotoxicity, and for these reasons they could further a promising approach to GBM [14,144]. 

A comprehensive review of the main clinical trials has been summed up in Table 1. 

## 7. Conclusions

Immunotherapy is a significant revolution in the management and care of tumors, including gliomas, and is rapidly evolving to meet the specific needs and challenges of different tumor types [14,22]. 

With immunotherapies, particularly immune checkpoint inhibitors, extraordinary responses have been observed in various tumor types outside of the CNS [55,145]. For glioblastoma, most attempts to incorporate immunotherapies into traditional chemo- and radiotherapy have so far been futile [43]. Their success in treating GBM has most likely been limited due to the tumor’s ability to evade immune surveillance by developing various immunoresistance mechanisms [146]. 

Indeed, strategies to overcome this immunoresistance include activation of the peritumoral inflammatory microenvironment; with a better understanding of the dynamics of the peritumoral microenvironment and improved preclinical tools, we can possibly develop more personalized and targeted treatments that could have a significant impact on patient survival [22]. The future direction of GBM therapy will include a combined approach that, in contrast to the inescapable current treatment modality of maximal resection followed by chemo- and radiotherapy, may combine a multifaceted immunotherapy treatment with the dual goals of directly killing tumor cells and activating the innate and adaptive immune response. 

## Figures and Tables

**Figure 1 biomedicines-11-01520-f001:**
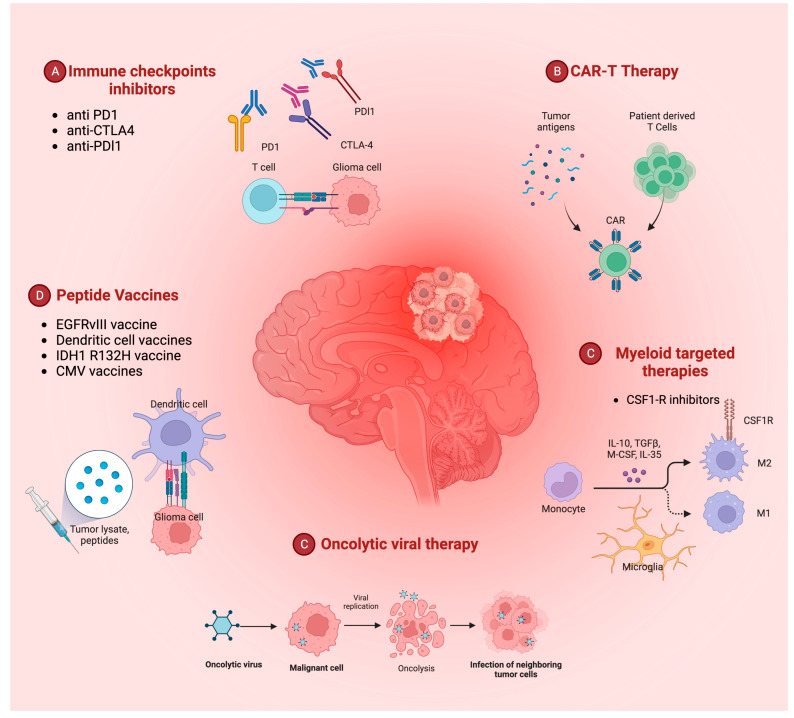
Overview of the current different immunotherapy-related therapeutic approaches to glioblastoma. PD1: programmed cell death protein 1; PDL1: programmed cell death protein ligand 1; CTLA4: cytotoxic T lymphocyte antigen; CMV: cytomegalovirus; IDH1: isocitrate dehydrogenase 1; EGFR: epidermal growth factor receptor; CAR: chimeric antigen receptor; CSF1-R: colony stimulating factor 1 receptor; TGF: transforming growth factor; IL: interleukin.

**Figure 2 biomedicines-11-01520-f002:**
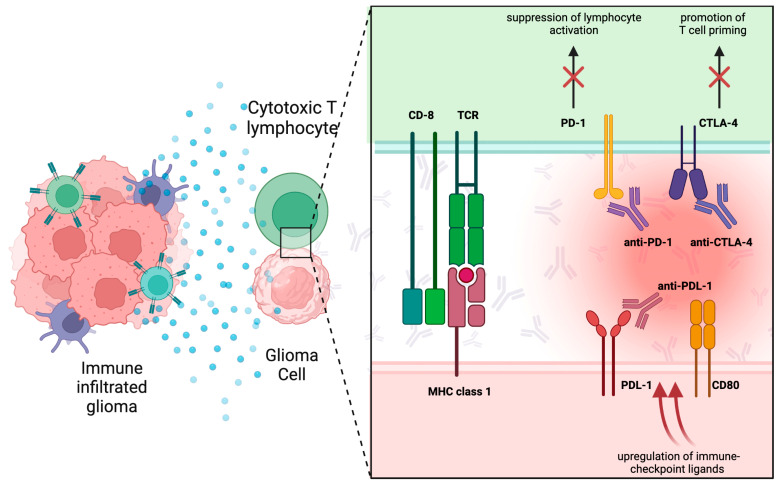
Schematic representation of the main immune checkpoint inhibitors. MHC: major histocompatibility complex; TCR: T cell receptor; CD: cluster of differentiation; PD1: programmed cell death protein 1: PDL1: programmed cell death protein ligand 1; CTLA4: cytotoxic T lymphocyte antigen.

**Figure 3 biomedicines-11-01520-f003:**
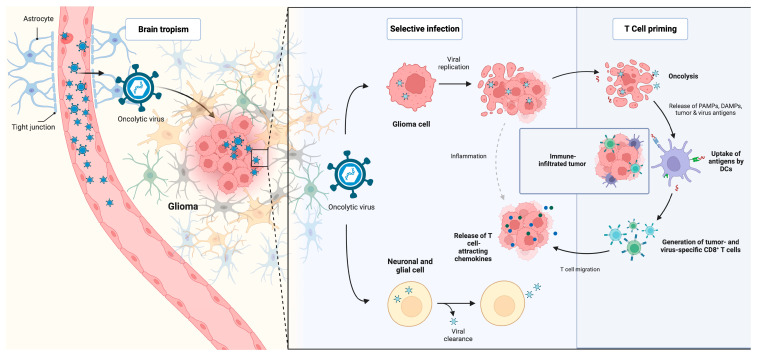
Mechanism of action and antitumoral activity of oncolytic viruses. DC: dendritic cells; CD: cluster of differentiation; PAMPs: pathogen-associated molecular patterns; DAMPs: damage-associated molecular patterns.

**Table 1 biomedicines-11-01520-t001:** Completed and ongoing clinical trials of immunotherapy for glioblastoma. CTLA4, Cytotoxic T-Lymphocyte Antigen 4. PD-1, Programmed cell death protein 1. OS, overall survival. PFS, progression-free survival. GBM, glioblastoma. NICs, Nanoscale immunoconjugates. mAbs monoclonal antibodies. TMZ, temozolomide. RT, radiation therapy. MGMT, (O[6]-methylguanine-DNA methyltransferase). EGFR, epidermal growth factor receptor. PEP-CMV, peptide vaccine derived from cytomegalovirus. DCs, dendritic cells. ATL, Autologous tumor lysate. TSC, Tumor Stem Cells. GAA, glioma-associated antigen. NK, natural killer. Treg, regulatory T lymphocyte. QoL, quality of life. pp65, phosphoprotein 65. CAR-T, chimeric antigen receptor T-cell. HER2, Human Epidermal Growth Factor Receptor 2.

Treatment	Target (s)	Type of Study	Year	Primary Endpoint	n°	Results	Identifier
NICs on Poly(β-L-malic acid) with covalently attached anti-CTLA 4 and anti PD-1 antibody	CTLA-4 PD-1	Murine	2019	OS of mice bearing intracranial GBM treated with free mAbs or NICs alone or in combination.		Significant improvement of OS in mice trated with checkpoint inhibitor mAb attached to NIC	
Nivolumab plus Ipilimumab	PD1 and CTLA-4	Phase III	2013	Effectiveness and Safety of Nivolumab Compared to Bevacizumab and of Nivolumab With or Without Ipilimumab in GBM Patients	529	Median OS was 9.8 months with nivolumab versus 10.0 months with bevacizumab	NCT02017717
Nivolumab	PD1	Phase III	2015	OS in Nivolumab compared to TMZ with RT for newly-diagnosed GBM	560	Median OS was 13.40 months with nivolumab versus 14.88 months in TMZ	NCT02617589 (concluded)
Nivolumab	PD1	Phase III	2016	OS in TMZ Plus RT combined with Nivolumab or placebo in newly diagnosed MGMT-Methylated GBM	716	Median PFS was 10.64 months with RT, TMZ plus Nivolumab versus 10.32 months in RT, TMZ Plus Placebo	NCT02667587 (ongoing)
VICTORI Rindopepimut	Vaccine anti-EGFR III	Phase I	2009	Rindopepimut toxicity in GBM patients with gross total resection and standard external beam RT	15	Minimal toxicity without symptoms of autoimmunity, without statistically significant improvement of outcome.	
ACTIVATE Rindopepimut	Vaccine anti-EGFR III	Phase II	2010	PFS and OS of vaccinated patients with newly diagnosed EGFRvIII-expressing GBM with minimal residual disease	35	OS and PFS of vaccinated patients were greater than that observed in a control group	NCT00643097
ACT II Rindopepimut	Vaccine anti-EGFR III and TMZ	Phase II	2011	If TMZ-induced lymphopenia with standard or intensified dose would enhance immune responses to the anti-EGFRIII-vaccine	22	Humoral and cellular vaccine-induced immune responses are more enhanced by a intensified TMZ dose than the standard TMZ dose	
ACT III Rindopepimut	Vaccine anti-EGFR III and TMZ	Phase II	2011	Efficacy and safety of Rindopepimut in EGFRvIII-positive GBM with gross total resection and no evidence of progression after RT and TMZ	65	Vaccine well-tolerated. Improved PFS and OS	NCT00458601
PERFORMANCE	PEP-CMV vaccination	Phase I	2016	Efficacy and safety of PEP-CMV vaccine	27	Vaccine generates an immune response No adverse events	NCT02864368 (terminated)
DCs vaccine	ATL-pulsed DCs vaccine	Phase I	2011	Vaccine safety and efficacy in inducing immunologic response in GBM after RT and TMZ.	10	Vaccinated patients with major immune response had improved survival, with no serious adverse events	
DCs vaccine	ATL-pulsed DCs vaccine versus GAA peptide-pulsed DCs vaccine	Phase I	2013	Comparison of safety, feasibility and immune responses of ATL-pulsed DC vaccine, with GAA peptide-pulsed DCs vaccine	34	More activated NK cells in GAA patients. Correlation between decreased Treg ratios (post/pre vaccination) and OS in both trials.	
DCs vaccine	TSC derived mRNA- Transfected DCs vaccine	Phase I Phase II	2009	Safety, immunological response, time to disease progression and survival time in vaccinated GBM patients	20	No adverse autoimmune events or other side effects. PFS was 2.9 times longer in vaccinated patients	NCT00846456 (completed)
DCs vaccine ICT-107	Autologous DCs pulsed with six synthetic peptide epitopes targeting GBM tumor/stem cell-associated antigens MAGE-1, HER-2, AIM-2, TRP-2, gp100, and IL13Rα2	Phase II	2017	ICT-107 tested efficacy, safety, QoL and immune response	124	No adverse autoimmune events. PFS significantly improved in ICT-107-treated patients with maintenance of QoL. HLA-A2 subgroup showed increased ICT-107 activity clinically and immunologically.	NCT01280552 (completed)
DCs vaccine	DC cells pulsed with CMV-pp65 RNA vaccine	Phase I	2017	Pp65-specific cellular responses and the effects on long-term PFS and OS	11	Long-term PFS (25.3 months) and OS (41.1 months) in vaccinated patients	
CAR-T therapy	Autologous anti-EGFRvIII CAR T cells	Phase I	2014	Safety and feasibility of CAR T-EGFRvIII	11	No incidence of cytokine-release syndrome or neurotoxicity. OS not affected by therapy	NCT02209376 (terminated)
CAR-T therapy	HER2-specific CAR-modified virus-specific T cells	Phase I	2019	Dose-Escalation Trial	16	Infusions well tolerated, with no dose-limiting toxic effects	NCT01109095 (completed)
Oncolytic viruses therapy	Recombinant oncolytic Polio/Rhinovirus PVSRIPO	Phase I	2021	Dose-finding and safety Study in recurrent GBM	61	Intratumoral reinfusion of PVSRIPO via CED is safe, and encouraging efficacy results have been observed	NCT01491893 (completed)
Oncolytic viruses DNX-2401 + Pembrolizumab	Genetically modified oncolytic adenovirus+ Anti-PD1	Phase II	2021	Objective response rate and OS	49	Not disclosed	NCT02798406 (completed)
Oncolytic viruses Toca 511	Vocimagene amiretrorepvec vector for a yeast cytosine deaminase gene which converts the prodrug Toca FC into the antimetabolite 5-fluorouracil	Phase I	2016	Safety, efficacy, and molecular profiling of Toca 511 OS	45	Excellent tolerability OS for recurrent high grade glioma was 13.6 months, statistically improved relative to an external control	

## Data Availability

Data sharing is not applicable to this article as no new data were created or analyzed in this study.

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
