# Peer review of "Updates in Glioblastoma Immunotherapy: An Overview of the Current Clinical and Translational Scenario"

_biomedicines, 2023, doi:10.3390/biomedicines11061520_

Round 1
Reviewer 1 Report
1. I did not find table 1, to which the authors refer in lines 156, 223, 505. I just wanted to write in the form of a comment that there is not enough table that would summarize the data on clinical trials.
2. The quality of the drawings is low, you need to make it clearer.
3. Many typos and negligence in the design of the text of the manuscript.
Author Response
We thank the reviewers for the opportunity and for their comments, which we try to address point by point.
Reviewer 1
- I did not find table 1, to which the authors refer in lines 156, 223, 505. I just wanted to write in the form of a comment that there is not enough table that would summarize the data on clinical trials.
The table is included in the supplementary materials, because it was too large to be contained in the main text without distorting the layout. We are aware that it cannot contain all studies on such a broad topic, so we have tried to select those of greatest interest to readers, which are also those we have cited in the main text. In particular, we selected clinical studies in humans, or with large populations, or targeting innovative molecules that had already demonstrated potential in vitro or in animal models.
- The quality of the drawings is low, you need to make it clearer.
We apologize for the inconvenience, the image quality is low because as per the guidelines for authors, the images were uploaded in the main text, and the quality was affected. Full resolution images will be uploaded in supplementary materials so that the editorial office can choose how to upload them in the best format.
- Many typos and negligence in the design of the text of the manuscript.
Again, we apologize for the inconvenience, we tried to correct as many typos as possible
Reviewer 2 Report
This manuscript reviews the use of immunotherapy in glioblastoma. The review is well written, it is easy to read, and to understand. A table summarizing the different approaches would help the reader, as there is a lot of information. Of note, figure 1 is also summarizing the main treatments, so it is up to the authors.
Comments:
(1) Could you please add a small paragraph describing the clinicopathological features of the most common types of brain tumors? The most common are metastatic, and meningioma. Of note, glioblastoma is the most common primary brain tumor. You may mention the differences between pediatric and adult types.
(2) Section present on lines 50 - 61, the immunotherapy is introduced. The fact that CNS is considered an immune privilege site is also mentioned. But, the lymphatic system that has been recently discovered is highlighted. Could you please describe with more details the immune microenvironment / host reponse of CNS? You could add figures, and/or tables with the most relevant cells and immune checkpoint molecules.
(3) In the Introduction, the glioblastoma histological classification could be described. The most recent WHO classification was published in 2021.
Louis DN, Perry A, Wesseling P, Brat DJ, Cree IA, Figarella-Branger D, Hawkins C, Ng HK, Pfister SM, Reifenberger G, Soffietti R, von Deimling A, Ellison DW. The 2021 WHO Classification of Tumors of the Central Nervous System: a summary. Neuro Oncol. 2021 Aug 2;23(8):1231-1251. doi: 10.1093/neuonc/noab106. PMID: 34185076; PMCID: PMC8328013. (4)
(4) Line 64. Could you please expand the difference between "cold" and "hot" tumors?
(5) Line 77. Could you please explain what produces IL10 in GBM?
(6) Line 85. Could you please explain the origin of the macrophages in GBM? Do they provide from the "CNS lymphatic system"?
(7) Dos the presence of GBM cells disrupts the immune privileged site mechanisms?
(8) Several immune checkpoint inhibitors are mentioned in the lines 101 - 109. Could you please add if all of them are agonistic or antagonistic antibodies?
(9) Line 216. Could you please add more general information regarding the design and mechanism of function of peptide vaccines?
(10) Line 251. Regarding IDH1. This marker relates to the WHO classification. This could be highlighted (if appropriate). Glioblastoma, IDH-wildtype, and adult-type diffuse gliomas IDH-mutant.
(11) Line 271. Is there any evidence that GBM is EBV-positive (EBER, LMP1, EBNA2, EBNA1, etc.)?
(12) Is there any successful clinical trial about CAR-T and GBM?
(13) In addition to the figures, a table summarizing the different approaches would help the reader as there are sections that are quite long.
(14) What are the prognostic factors of GMB (in adults)? Age and performance status? What molecular alterations also carry prognostic value (IDH, 1p, 19q, MGMT).
(15) There are many immune checkpoint markers. But, from a histopathologic point of view, do GBM express all of them at protein level? If not express, should the immunecheckpoint approach not be used?
Author Response
Reviewer 2
This manuscript reviews the use of immunotherapy in glioblastoma. The review is well written, it is easy to read, and to understand. A table summarizing the different approaches would help the reader, as there is a lot of information. Of note, figure 1 is also summarizing the main treatments, so it is up to the authors.
Comments:
(1) Could you please add a small paragraph describing the clinicopathological features of the most common types of brain tumors? The most common are metastatic, and meningioma. Of note, glioblastoma is the most common primary brain tumor. You may mention the differences between pediatric and adult types.
I think adding a paragraph on the main features of brain tumors may be redundant in a review that is itself very long. As suggested we added the fact that they are the most common primary tumor. Everything written in this review refers to adult glioblastomas, we have added this specific in the introduction
(2) Section present on lines 50 - 61, the immunotherapy is introduced. The fact that CNS is considered an immune privilege site is also mentioned. But, the lymphatic system that has been recently discovered is highlighted. Could you please describe with more details the immune microenvironment / host reponse of CNS? You could add figures, and/or tables with the most relevant cells and immune checkpoint molecules.
We added a paragraph on cns- microenviroment interaction as suggested, focusing on the creation of the tumor inflammatory microenvironment
(3) In the Introduction, the glioblastoma histological classification could be described. The most recent WHO classification was published in 2021.
Louis DN, Perry A, Wesseling P, Brat DJ, Cree IA, Figarella-Branger D, Hawkins C, Ng HK, Pfister SM, Reifenberger G, Soffietti R, von Deimling A, Ellison DW. The 2021 WHO Classification of Tumors of the Central Nervous System: a summary. Neuro Oncol. 2021 Aug 2;23(8):1231-1251. doi: 10.1093/neuonc/noab106. PMID: 34185076; PMCID: PMC8328013. (4)
We apologize for the missing reference, which was added in the introduction section
(4) Line 64. Could you please expand the difference between "cold" and "hot" tumors?
the difference between hot and cold tumors was added in the quoted lines
(5) Line 77. Could you please explain what produces IL10 in GBM?
As added in the main text, il-10 is secreted by both tumor-associated macrophages and tumor cells themselves
(6) Line 85. Could you please explain the origin of the macrophages in GBM? Do they provide from the "CNS lymphatic system"?
Between lines 88 and 92 we added a short explanatory paragraph on the tumor microenvironment and the origin of macrophages. They do not only originate from the lymphatic system, but this has been a particularly studied pathway in recent years
(7) Dos the presence of GBM cells disrupts the immune privileged site mechanisms?
Partially yes, but non entirely: in the sense that the damage caused by GBM causes a disruption of the blood-brain barrier, consequently causinginflammation, but on the other hand GBM promotes anti-inflammatory pathway signalling itself, which counteracts this basic mechanism of BBB destruction
(8) Several immune checkpoint inhibitors are mentioned in the lines 101 - 109. Could you please add if all of them are agonistic or antagonistic antibodies?
They all are antagonists, that is, they have the function of binding to the target preventing its activation. None of them promote its activation. This concept is more fully specified in the appropriate section 3. Immune Checkpoint Inhibitors
(9) Line 216. Could you please add more general information regarding the design and mechanism of function of peptide vaccines?
A brief introduction has been added at the beginning of the Peptide Vaccine section
(10) Line 251. Regarding IDH1. This marker relates to the WHO classification. This could be highlighted (if appropriate). Glioblastoma, IDH-wildtype, and adult-type diffuse gliomas IDH-mutant.
yes, mutation correlates with who classification and can be a harbinger of misunderstanding. In fact, according to WHO 2016 there were IDH-mutant GB, whereas nowadays these same tumors are classified as grade 4 IDH-mutant astrocytomas. We tried to clarify this point at the beginning of the section on IDH vaccine.
(11) Line 271. Is there any evidence that GBM is EBV-positive (EBER, LMP1, EBNA2, EBNA1, etc.)?
yes, there is some evidence about the presence of GBM with EBV-positivity, but it's not so straight forward. In the only study specifically conducted in this regard, 20% of patients showed EBV positivity.
PMID: 29732319; PMID: 31501045
(12) Is there any successful clinical trial about CAR-T and GBM?
There are still no results demonstrating an efficacy of CAR therapy in increasing OS or PFS in GBM patients. Clinical trials in humans for now have focused on establishing its safety in patients after multiple lines of treatment, and on highlighting its efficacy in immune activation.
(13) In addition to the figures, a table summarizing the different approaches would help the reader as there are sections that are quite long.
As requested in the initial comment section, we understand the reviewer's request in wanting to include a summary table of the various treatment approaches. We preferred the figure because it was of more immediate impact and understanding, whereas the table was in our opinion less useful in generically distinguishing the various therapies, and we used it to summarize the clinical trials currently underway by highlighting their various features (see supplemetary materials). We don’t want to be redundant summing more or less the same concepts in two forms, in a review which is quite long itself. However, if the reviewers see it as advisable surely we can generate such a table.
(14) What are the prognostic factors of GBM (in adults)? Age and performance status? What molecular alterations also carry prognostic value (IDH, 1p, 19q, MGMT).
The prognostic factors that affect the outcome of GBM patients are many, going through macroscopic factors related to the tumor itself, such as location in the brain, volume, radiological features; clinical factors such as the presence of seizures, neurological deficits, and performance status; and histopathological factors, such as the mutations you mentioned. In particular, the IDH mutation and the 1p/19q codeletion have such a significant impact that in the new WHO classification they are sufficient element to no longer classify the tumor as glioblastoma. MGMT promoter methylation, and the degree of methylation itself, on the other hand, has remained an independent positive prognostic factor that correlates with increased response to adjuvant therapy. We decided not to include this information in the text because since the review is already very long, this aspect might be of less interest to a reader seeking information specifically on immunotherapy.
(15) There are many immune checkpoint markers. But, from a histopathologic point of view, do GBM express all of them at protein level? If not express, should the immunecheckpoint approach not be used?
Yes, all of these immune checkpoints molecules are expressed at a protein level in GBM patients, but, as detailed in the text in the immune checkpoints section, the main obstacle to this approach is the fact that gbm is a cold tumor, and the inflammatory response has to be somehow elicited in order to think that these strategies have efficacy. The second remarkable obstacle is that BBB prevents currently available drugs from having sufficient concentration in brain tissue.